# Causal Shapley Values: Exploiting Causal Knowledge to Explain Individual Predictions of Complex Models

**Tom Heskes**
Radboud University
tom.heskes@ru.nl

**Evi Sijben**
Machine2Learn
evisijben@gmail.com

**Ioan Gabriel Bucur**
Radboud University
g.bucur@cs.ru.nl

**Tom Claassen**
Radboud University
t.claassen@science.ru.nl

## Abstract

Shapley values underlie one of the most popular model-agnostic methods within explainable artificial intelligence. These values are designed to attribute the difference between a model's prediction and an average baseline to the different features used as input to the model. Being based on solid game-theoretic principles, Shapley values uniquely satisfy several desirable properties, which is why they are increasingly used to explain the predictions of possibly complex and highly non-linear machine learning models. Shapley values are well calibrated to a user's intuition when features are independent, but may lead to undesirable, counterintuitive explanations when the independence assumption is violated.

In this paper, we propose a novel framework for computing Shapley values that generalizes recent work that aims to circumvent the independence assumption. By employing Pearl's *do*-calculus, we show how these 'causal' Shapley values can be derived for general causal graphs without sacrificing any of their desirable properties. Moreover, causal Shapley values enable us to separate the contribution of direct and indirect effects. We provide a practical implementation for computing causal Shapley values based on causal chain graphs when only partial information is available and illustrate their utility on a real-world example.

## 1 Introduction

Complex machine learning models like deep neural networks and ensemble methods like random forest and gradient boosting machines may well outperform simpler approaches such as linear regression or single decision trees, but are noticeably harder to interpret. This can raise practical, ethical, and legal issues, most notably when applied in critical systems, e.g., for medical diagnosis or autonomous driving. The field of explainable AI aims to address these issues by enhancing the interpretability of complex machine learning models.

The Shapley-value approach has quickly become one of the most popular model-agnostic methods within explainable AI. It can provide local explanations, attributing changes in predictions for individual data points to the model's features, that can be combined to obtain better global understanding of the model structure [17]. Shapley values are based on a principled mathematical foundation [27] and satisfy various desiderata (see also Section 2). They have been applied for explaining statistical and machine learning models for quite some time, see e.g., [15, 31]. Recent interests have been triggered by Lundberg and Lee's breakthrough paper [19] that introduces efficient computational procedures and unifies Shapley values and other popular local model-agnostic approaches such as LIME [26].

Humans have a strong tendency to reason about their environment in causal terms [28], where explanation and causation are intimately related: explanations often appeal to causes, and causal claims often answer questions about why or how something occurred [16]. The specific domain of causal responsibility studies how people attribute an effect to one or more causes, all of which may have contributed to the observed effect [29]. Causal attributions by humans strongly depend on a subject's understanding of the generative model that explains how different causes lead to the effect, for which the relations between these causes are essential [7].

Most explanation methods, however, tend to ignore such relations and act as if features are independent. Even so-called counterfactual approaches, that strongly rely on a causal intuition, make this simplifying assumption (e.g., [33]) and ignore that, in the real world, a change in one input feature may cause a change in another. This independence assumption also underlies early Shapley-based approaches, such as [31, 3], and is made explicit as an approximation for computational reasons in [19]. We will refer to these as *marginal* Shapley values.

Aas et al. [1] argue and illustrate that marginal Shapley values may lead to incorrect explanations when features are highly correlated, motivating what we will refer to as *conditional* Shapley values. Janzing et al. [8], following [3], discuss a causal interpretation of Shapley values, in which they replace conventional conditioning by observation with conditioning by intervention, as in Pearl's *do*-calculus [24]. They argue that, when the goal is to causally explain the prediction *algorithm*, the inputs of this algorithm can be formally distinguished from the features in the real world and 'interventional' Shapley values simplify to marginal Shapley values. This argument is also picked up by [17] when implementing interventional Tree SHAP. Going in a different direction, Frye et al. [6] propose *asymmetric* Shapley values as a way to incorporate causal knowledge in the real world by restricting the possible permutations of the features when computing the Shapley values to those consistent with a (partial) causal ordering. In line with [1], they then apply conventional conditioning by observation to make sure that the explanations respect the data manifold.

The main contributions of our paper are as follows. (1) We derive *causal* Shapley values that explain the total effect of features on the prediction, taking into account their causal relationships. This makes them principally different from marginal and conditional Shapley values. At the same time, compared to asymmetric Shapley values, causal Shapley values provide a more direct and robust way to incorporate causal knowledge. (2) Our method allows for further insights into feature relevance by separating out the total causal effect into a direct and indirect contribution. (3) Making use of causal chain graphs [13], we propose a practical approach for computing causal Shapley values and illustrate this on a real-world example.

## 2    Causal Shapley values

In this section, we will introduce causal Shapley values and contrast them to other approaches. We assume that we are given a machine learning model $f(\cdot)$ that can generate predictions for any feature vector $\mathbf{x}$. Our goal is to provide an explanation for an individual prediction $f(\mathbf{x})$, that takes into account the causal relationships between the features in the real world.

Attribution methods, with Shapley values as their most prominent example, provide a local explanation of individual predictions by attributing the difference between $f(\mathbf{x})$ and a baseline $f_0$ to the different features $i \in N$ with $N = \{1, \ldots, n\}$ and $n$ the number of features:

$$f(\mathbf{x}) = f_0 + \sum_{i=1}^{n} \phi_i \, , \tag{1}$$

where $\phi_i$ is the contribution of feature $i$ to the prediction $f(\mathbf{x})$. For the baseline $f_0$ we will take the average prediction $f_0 = \mathbb{E}f(\mathbf{X})$ with expectation taken over the observed data distribution $P(\mathbf{X})$. Equation (1) is referred to as the *efficiency property* [27], which appears to be a sensible desideratum for any attribution method and we therefore take here as our starting point.

To go from knowing none of the feature values, as for $f_0$, to knowing all feature values, as for $f(\mathbf{x})$, we can add feature values one by one, actively setting the features to their values in a particular order $\pi$. We define the contribution of feature $i$ given permutation $\pi$ as the difference in value function before and after setting the feature to its value:

$$\phi_i(\pi) = v(\{j : j \preceq_\pi i\}) - v(\{j : j \prec_\pi i\}) \, , \tag{2}$$

with $j \prec_\pi i$ if $j$ precedes $i$ in the permutation $\pi$ and where we choose the value function

$$v(S) = \mathbb{E}\left[f(\mathbf{X})|do(\mathbf{X}_S = \mathbf{x}_S)\right] = \int d\mathbf{X}_{\bar{S}}\, P(\mathbf{X}_{\bar{S}}|do(\mathbf{X}_S = \mathbf{x}_S))f(\mathbf{X}_{\bar{S}}, \mathbf{x}_S)\,. \qquad (3)$$

Here $S$ is the subset of 'in-coalition' indices with known feature values $\mathbf{x}_S$. To compute the expectation, we average over the 'out-of-coalition' or dropped features $\mathbf{X}_{\bar{S}}$ with $\bar{S} = N \setminus S$, the complement of $S$. To explicitly take into account possible causal relationships between the 'in-coalition' features and the 'out-of-coalition' features, we propose to condition 'by intervention' for which we resort to Pearl's *do*-calculus [23]. In words, the contribution $\phi_i(\pi)$ now measures the relevance of feature $i$ through the (average) prediction obtained if we actively set feature $i$ to its value $x_i$ compared to (the counterfactual situation of) not knowing its value.

Since the sum over features $i$ in (2) is telescoping, the efficiency property (1) holds for any permutation $\pi$. Therefore, for any distribution over permutations $w(\pi)$ with $\sum_\pi w(\pi) = 1$, the contributions

$$\phi_i = \sum_\pi w(\pi)\phi_i(\pi) \qquad (4)$$

still satisfy (1). An obvious choice would be to take a uniform distribution $w(\pi) = 1/n!$. We then arrive at (with shorthand $i$ for the singleton $\{i\}$):

$$\phi_i = \sum_{S \subseteq N \setminus i} \frac{|S|!(n - |S| - 1)!}{n!} \left[v(S \cup i) - v(S)\right]\,.$$

Besides efficiency, the Shapley values uniquely satisfy three other desirable properties [27].

**Linearity:** for two value functions $v_1$ and $v_2$, we have $\phi_i(\alpha_1 v_1 + \alpha_2 v_2) = \alpha_1 \phi_i(v_1) + \alpha_2 \phi_i(v_2)$. This guarantees that the Shapley value of a linear ensemble of models is a linear combination of the individual models' Shapley values.

**Null player (dummy):** if $v(S \cup i) = v(S)$ for all $S \subseteq N \setminus i$, then $\phi_i = 0$. A feature that never contributes to the prediction (directly nor indirectly, see below) receives zero Shapley value.

**Symmetry:** if $v(S \cup i) = v(S \cup j)$ for all $S \subseteq N \setminus \{i, j\}$, then $\phi_i = \phi_j$. Symmetry in this sense holds for marginal, conditional, and causal Shapley values alike.

Note that symmetry here is defined w.r.t. to the contributions $\phi_i$, not the function values $f(\mathbf{x})$. As discussed in [8] (Section 3), conditioning by observation or intervention then does not break the symmetry property. For a non-uniform distribution of permutations as in [6], symmetry is lost, but efficiency, linearity, and null player still apply.

Replacing conditioning by intervention with conventional conditioning by observation, i.e., averaging over $P(\mathbf{X}_{\bar{S}}|\mathbf{x}_S)$ instead of $P(\mathbf{X}_{\bar{S}}|do(\mathbf{X}_S = \mathbf{x}_S))$ in (3), we arrive at the conditional Shapley values of [1, 18]. A third option is to ignore the feature values $\mathbf{x}_S$ and take the unconditional, marginal distribution $P(\mathbf{X}_{\bar{S}})$, which leads to the marginal Shapley values.

Up until here, our active interventional interpretation of Shapley values coincides with that in [3, 8, 17]. However, from here on Janzing et al. [8] choose to ignore any dependencies between the features in the real world, by formally distinguishing between true features (corresponding to one of the data points) and the features plugged as input into the model. This leads to the conclusion that, in our notation, $P(\mathbf{X}_{\bar{S}}|do(\mathbf{X}_S = \mathbf{x}_S)) = P(\mathbf{X}_{\bar{S}})$ for any subset $S$. As a result, any expectation under conditioning by intervention collapses to a marginal expectation and, in the interpretation of [3, 8, 17], interventional Shapley values simplify to marginal Shapley values. As we will see below, marginal Shapley values can only represent direct effects, which makes that 'root causes' with strong indirect effects (e.g. genetic markers) are ignored in the attribution, which is quite different from how humans tend to attribute causes [29]. In this paper, we choose not to make this distinction between the features in the real world and the inputs of the prediction model, but to explicitly take into account the causal relationships between the data in the real world to enhance the explanations. Since the term 'interventional' Shapley values has been coined for causal explanations of the prediction algorithm, ignoring causal relationships between the features in the real world, we will use the term 'causal' Shapley values for those that do attempt to incorporate these relationships using Pearl's *do*-calculus.

The asymmetric Shapley values introduced in [6] (see also these proceedings) have the same objective: enhancing the explanation of the Shapley values by incorporating causal knowledge about the features

in the real world. In [6], this knowledge is incorporated by choosing $w(\pi) \neq 0$ in (4) only for those permutations $\pi$ that are consistent with the causal structure between the features, i.e., are such that a known causal ancestor always precedes its descendants. On top of this, Frey et al. [6] apply standard conditioning by observation. In this paper we show that there is no need to resort to asymmetric Shapley values to incorporate causal knowledge: applying conditioning by intervention instead of conditioning by observation is sufficient. Choosing asymmetric Shapley values instead of symmetric ones can be considered orthogonal to choosing conditioning by observation versus conditioning by intervention. We will therefore refer to the approach of [6] as *asymmetric conditional* Shapley values, to contrast them with *asymmetric causal* Shapley values that implement both ideas.

## 3 Decomposing Shapley values into direct and indirect effects

Our causal interpretation allows us to distinguish between direct and indirect effects of each feature on a model's prediction. This decomposition then also helps to understand the difference between marginal, symmetric, and asymmetric Shapley values. Going back to the contribution $\phi_i(\pi)$ for a permutation $\pi$ and feature $i$ in (2) and using shorthand notation $\underline{S} = \{j : j \prec_\pi i\}$ and $\bar{S} = \{j : j \succ_\pi i\}$, we can decompose the total effect for this permutation into a direct and an indirect effect:

$$
\begin{aligned}
\phi_i(\pi) &= \mathbb{E}[f(\mathbf{X}_{\bar{S}}, \mathbf{x}_{\underline{S} \cup i}) | do(\mathbf{X}_{\underline{S} \cup i} = \mathbf{x}_{\underline{S} \cup i})] - \mathbb{E}[f(\mathbf{X}_{\bar{S} \cup i}, \mathbf{x}_{\underline{S}}) | do(\mathbf{X}_{\underline{S}} = \mathbf{x}_{\underline{S}})] & \text{(total effect)} \\
&= \mathbb{E}[f(\mathbf{X}_{\bar{S}}, \mathbf{x}_{\underline{S} \cup i}) | do(\mathbf{X}_{\underline{S}} = \mathbf{x}_{\underline{S}})] - \mathbb{E}[f(\mathbf{X}_{\bar{S} \cup i}, \mathbf{x}_{\underline{S}}) | do(\mathbf{X}_{\underline{S}} = \mathbf{x}_{\underline{S}})] + & \text{(direct effect)} \\
&\quad \mathbb{E}[f(\mathbf{X}_{\bar{S}}, \mathbf{x}_{\underline{S} \cup i}) | do(\mathbf{X}_{\underline{S} \cup i} = \mathbf{x}_{\underline{S} \cup i})] - \mathbb{E}[f(\mathbf{X}_{\bar{S}}, \mathbf{x}_{\underline{S} \cup i}) | do(\mathbf{X}_{\underline{S}} = \mathbf{x}_{\underline{S}})] & \text{(indirect effect)}
\end{aligned}
$$
(5)

The direct effect measures the expected change in prediction when the stochastic feature $X_i$ is replaced by its feature value $x_i$, without changing the distribution of the other 'out-of-coalition' features. The indirect effect measures the difference in expectation when the distribution of the other 'out-of-coalition' features changes due to the additional intervention $do(X_i = x_i)$. The direct and indirect parts of Shapley values can then be computed as in (4): by taking a, possibly weighted, average over all permutations. Conditional Shapley values can be decomposed similarly by replacing conditioning by intervention with conditioning by observation in (5). For marginal Shapley values, there is no conditioning and hence no indirect effect: by construction marginal Shapley values can only represent direct effects. We will make use of this decomposition in the next section to clarify how different causal structures lead to different Shapley values.

## 4 Shapley values for different causal structures

To illustrate the difference between the various Shapley values, we consider four causal models on two features. They are constructed such that they have the same $P(\mathbf{X})$, with $\mathbb{E}[X_2|x_1] = \alpha x_1$ and $\mathbb{E}[X_1] = \mathbb{E}[X_2] = 0$, but with different causal explanations for the dependency between $X_1$ and $X_2$. In the causal chain $X_1$ could, for example, represent season, $X_2$ temperature, and $Y$ bike rental. The fork inverts the arrow between $X_1$ and $X_2$, where now $Y$ may represent hotel occupation, $X_2$ season, and $X_1$ temperature. In the chain and the fork, different data points correspond to different days. For the confounder and the cycle, $X_1$ and $X_2$ may represent obesity and sleep apnea, respectively, and $Y$ hours of sleep. The confounder model implements the assumption that obesity and sleep apnea have a common confounder $Z$, e.g., some genetic predisposition. The cycle, on the other hand, represents the more common assumption that there is a reciprocal effect, with obesity affecting sleep apnea and vice versa [22]. In the confounder and the cycle, different data points correspond to different subjects. We assume to have trained a linear model $f(x_1, x_2)$ that happens to largely, or even completely to simplify the formulas, ignore the first feature, and boils down to the prediction function $f(x_1, x_2) = \beta x_2$. Figure 1 shows the explanations provided by the various Shapley values for each of the causal models in this extreme situation. Derivations can be found in the supplement.

Since in all cases there is no direct link between $X_1$ and the prediction, the direct effect of $X_1$ is always equal to zero. Similarly, any indirect effect of $X_2$ can only go through $X_1$ and hence must also be equal to zero. So, all we can expect is a direct effect from $X_2$, proportional to $\beta$, and an indirect effect from $X_1$ through $X_1$, proportional to $\alpha$ times $\beta$. Because of the sufficiency property, the direct and indirect effect always add up to the output $\beta x_2$. This makes that, for all different combinations of causal structures and types of Shapley values, we end up with just three different explanation patterns, referred to as *D*, *E*, and *R* in Figure 1.

| | $D$ direct | $D$ indirect | $E$ direct | $E$ indirect | $R$ direct | $R$ indirect |
|---|---|---|---|---|---|---|
| $\phi_1$ | $0$ | $0$ | $0$ | $\frac{1}{2}\beta\alpha x_1$ | $0$ | $\beta\alpha x_1$ |
| $\phi_2$ | $\beta x_2$ | $0$ | $\beta x_2 - \frac{1}{2}\beta\alpha x_1$ | $0$ | $\beta x_2 - \beta\alpha x_1$ | $0$ |

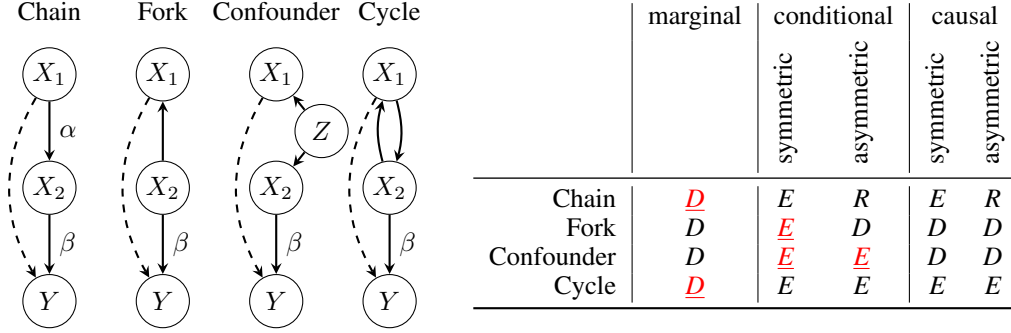

| | marginal | conditional symmetric | conditional asymmetric | causal symmetric | causal asymmetric |
|---|---|---|---|---|---|
| Chain | *D* | $E$ | $R$ | $E$ | $R$ |
| Fork | $D$ | *E* | $D$ | $D$ | $D$ |
| Confounder | $D$ | *E* | *E* | $D$ | $D$ |
| Cycle | *D* | $E$ | $E$ | $E$ | $E$ |

Figure 1: Direct and indirect Shapley values for four causal models with the same observational distribution over features (such that $\mathbb{E}[X_1] = \mathbb{E}[X_2] = 0$ and $\mathbb{E}[X_2|x_1] = \alpha x_1$), yet a different causal structure. We assume a linear model that happens to ignore the first feature: $f(x_1, x_2) = \beta x_2$. The bottom table gives for each of the four causal models on the left the marginal, conditional, and causal Shapley values, where the latter two are further split up in symmetric and asymmetric. Each letter in the bottom table corresponds to one of the patterns of direct and indirect effects detailed in the top table: 'direct' (*D*, only direct effects), 'evenly split' (*E*, credit for an indirect effect split evenly between the features), and 'root cause' (*R*, all credit for the indirect effect goes to the root cause). Shapley values that, we argue, do not provide a proper causal explanation, are underlined and indicated in red.

To argue which explanations make sense, we call upon classical norm theory [9]. It states that humans, when asked for an explanation of an effect, contrast the actual observation with a counterfactual, more normal alternative. What is considered normal, depends on the context. Shapley values can be given the same interpretation [20]: they measure the difference in prediction between knowing and not knowing the value of a particular feature, where the choice of what's normal translates to the choice of the reference distribution to average over when the feature value is still unknown.

In this perspective, marginal Shapley values as in [3, 8, 17] correspond to a very simplistic, counterintuitive interpretation of what's normal. Consider for example the case of the chain, with $X_1$ representing season, $X_2$ temperature, and $Y$ bike rental, and two days with the same temperature of 13 degrees Celsius, one in fall and another in winter. Marginal Shapley values end up with the same explanation for the predicted bike rental on both days, ignoring that the temperature in winter is higher than normal for the time of year and in fall lower. Just like marginal Shapley values, symmetric conditional Shapley values as in [1] do not distinguish between any of the four causal structures. They do take into account the dependency between the two features, but then fail to acknowledge that an *intervention* on feature $X_1$ in the fork and the confounder, does not change the distribution of $X_2$.

For the confounder and the cycle, asymmetric Shapley values put $X_1$ and $X_2$ on an equal footing and then coincide with their symmetric counterparts. Asymmetric *conditional* Shapley values from [6] have no means to distinguish between the cycle and the confounder, unrealistically assigning credit to $X_1$ in the latter case. Asymmetric and symmetric *causal* Shapley values do correctly treat the cycle and confounder cases.

In the case of a chain, asymmetric and symmetric causal Shapley values provide different explanations. Which explanation is to be preferred may well depend on the context. In our bike rental example, asymmetric Shapley values first give full credit to season for the indirect effect (here $\alpha\beta x_1$), subtracting this from the direct effect of the temperature to fulfill the sufficiency property ($\beta x_2 - \alpha\beta x_1$). Symmetric causal Shapley values consider both contexts – one in which season is intervened upon before temperature, and one in which temperature is intervened upon before season –

and then average over the results in these two contexts. This symmetric strategy appears to better appeal to the theory dating back to [14], that humans sample over different possible scenarios (here: different orderings of the features) to judge causation. However, when dealing with a temporal chain of events, alternative theories (see e.g. [30]) suggest that humans have a tendency to attribute credit or blame foremost to the root cause, which seems closer in spirit to the explanation provided by asymmetric causal Shapley values.

By dropping the symmetry property, asymmetric Shapley values do pay a price: they are sensitive to the insertion of causal links with zero strength. As an example, consider a neural network trained to perfectly predict the XOR function on two binary variables $X_1$ and $X_2$. With a uniform distribution over all features and no further assumption w.r.t. the causal ordering of $X_1$ and $X_2$, the Shapley values are $\phi_1 = \phi_2 = 1/4$ when the prediction $f$ equals 1, and $\phi_1 = \phi_2 = -1/4$ for $f = 0$: completely symmetric. If we now assume that $X_1$ preceeds $X_2$ (and a causal strength of 0 to maintain the uniform distribution over features), all Shapley values stay the same, except for the asymmetric ones: these suddenly jump to $\phi_1 = 0$ and $\phi_2 = 1/2$ for $f = 1$, and $\phi_1 = 0$ and $\phi_2 = -1/2$ for $f = 0$. More details on this instability of asymmetric Shapley values can be found in the supplement, where we compare Shapley values of trained neural networks for varying causal strengths.

To summarize, unlike marginal and (both symmetric and asymmetric) conditional Shapley values, causal Shapley values provide sensible explanations that incorporate causal relationships in the real world. Asymmetric causal Shapley values may be preferable over symmetric ones when causality derives from a clear temporal order, whereas symmetric Shapley values have the advantage of being much less sensitive to model misspecifications.

## 5   A practical implementation with causal chain graphs

In the ideal situation, a practitioner has access to a fully specified causal model that can be plugged in (3) to compute or sample from every interventional probability of interest. In practice, such a requirement is hardly realistic. In fact, even if a practitioner could specify a complete causal structure (including potential confounding) and has full access to the observational probability $P(\mathbf{X})$, not every causal query need be identifiable (see e.g., [24]). Furthermore, requiring so much prior knowledge could be detrimental to the method's general applicability. In this section, we describe a pragmatic approach that is applicable when we have access to a (partial) causal ordering plus a bit of additional information to distinguish confounders from mutual interactions, and a training set to estimate (relevant parameters of) $P(\mathbf{X})$. Our approach is inspired by [6], but extends it in various aspects: it provides a formalization in terms of causal chain graphs, applies to both symmetric and asymmetric Shapley values, and correctly distinguishes between dependencies that are due to confounding and mutual interactions.

In the special case that a complete causal ordering of the features can be given and that all causal relationships are unconfounded, $P(\mathbf{X})$ satisfies the Markov properties associated with a directed acyclic graph (DAG) and can be written in the form

$$P(\mathbf{X}) = \prod_{j \in N} P(X_j | \mathbf{X}_{pa(j)}) \,,$$

with $pa(j)$ the parents of node $j$. With no further conditional independences, the parents of $j$ are all nodes that precede $j$ in the causal ordering. For causal DAGs, we have the interventional formula [13]:

$$P(\mathbf{X}_{\bar{S}} | do(\mathbf{X}_S = \mathbf{x}_S)) = \prod_{j \in \bar{S}} P(X_j | \mathbf{X}_{pa(j) \cap \bar{S}}, \mathbf{x}_{pa(j) \cap S}) \,, \tag{6}$$

with $pa(j) \cap T$ the parents of $j$ that are also part of subset $T$. The interventional formula can be used to answer any causal query of interest.

When we cannot give a complete ordering between the individual variables, but still a partial ordering, causal chain graphs [13] come to the rescue. A causal chain graph has directed and undirected edges. All features that are treated on an equal footing are linked together with undirected edges and become part of the same chain component. Edges between chain components are directed and represent causal relationships. See Figure 2 for an illustration of the procedure. The probability distribution

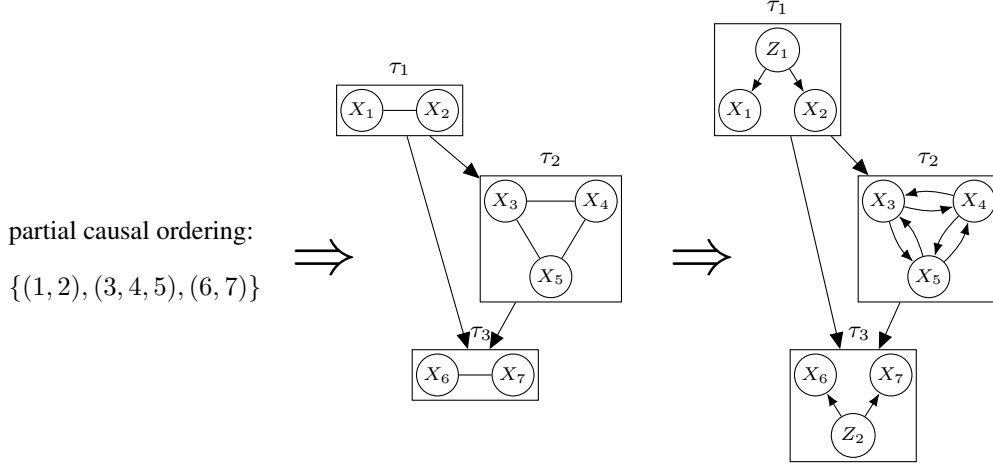

Figure 2: From partial ordering to causal chain graph. Features on an equal footing are combined into a fully connected chain component. How to handle interventions within each component depends on the generative process that best explains the (surplus) dependencies. In this example, the dependencies in chain components $\tau_1$ and $\tau_3$ are assumed to be the result of a common confounder, and those in $\tau_2$ of mutual interactions.

$P(\mathbf{X})$ in a chain graph factorizes as a "DAG of chain components":

$$P(\mathbf{X}) = \prod_{\tau \in \mathcal{T}} P(\mathbf{X}_\tau | \mathbf{X}_{pa(\tau)}) ,$$

with each $\tau$ a chain component, consisting of all features that are treated on an equal footing.

How to compute the effect of an intervention depends on the interpretation of the generative process leading to the (surplus) dependencies between features within each component. If we assume that these are the consequence of marginalizing out a common confounder, intervention on a particular feature will break the dependency with the other features. We will refer to the set of chain components for which this applies as $\mathcal{T}_{\text{confounding}}$. The undirected part can also correspond to the equilibrium distribution of a dynamic process resulting from interactions between the variables within a component [13]. In this case, setting the value of a feature does affect the distribution of the variables within the same component. We refer to these sets of components as $\mathcal{T}_{\overline{\text{confounding}}}$.

Any expectation by intervention needed to compute the causal Shapley values can be translated to an expectation by observation, by making use of the following theorem (see the supplement for a more detailed proof and some corollaries linking back to other types of Shapley values as special cases).

**Theorem 1.** *For causal chain graphs, we have the interventional formula*

$$P(\mathbf{X}_{\bar{S}}|do(\mathbf{X}_S = \mathbf{x}_S)) = \prod_{\tau \in \mathcal{T}_{\text{confounding}}} P(\mathbf{X}_{\tau \cap \bar{S}}|\mathbf{X}_{pa(\tau) \cap \bar{S}}, \mathbf{x}_{pa(\tau) \cap S}) \times$$

$$\prod_{\tau \in \mathcal{T}_{\overline{\text{confounding}}}} P(\mathbf{X}_{\tau \cap \bar{S}}|\mathbf{X}_{pa(\tau) \cap \bar{S}}, \mathbf{x}_{pa(\tau) \cap S}, \mathbf{x}_{\tau \cap S}) . \qquad (7)$$

*Proof.*

$$P(\mathbf{X}_{\bar{S}}|do(\mathbf{X}_S = \mathbf{x}_S)) \stackrel{(1)}{=} \prod_{\tau \in \mathcal{T}} P(\mathbf{X}_{\tau \cap \bar{S}}|\mathbf{X}_{pa(\tau) \cap \bar{S}}, do(\mathbf{X}_S = \mathbf{x}_S))$$

$$\stackrel{(3)}{=} \prod_{\tau \in \mathcal{T}} P(\mathbf{X}_{\tau \cap \bar{S}}|\mathbf{X}_{pa(\tau) \cap \bar{S}}, do(\mathbf{X}_{pa(\tau) \cap S} = \mathbf{x}_{pa(\tau) \cap S}), do(\mathbf{X}_{\tau \cap S} = \mathbf{x}_{\tau \cap S}))$$

$$\stackrel{(2)}{=} \prod_{\tau \in \mathcal{T}} P(\mathbf{X}_{\tau \cap \bar{S}}|\mathbf{X}_{pa(\tau) \cap \bar{S}}, \mathbf{x}_{pa(\tau) \cap S}, do(\mathbf{X}_{\tau \cap S} = \mathbf{x}_{\tau \cap S})) ,$$

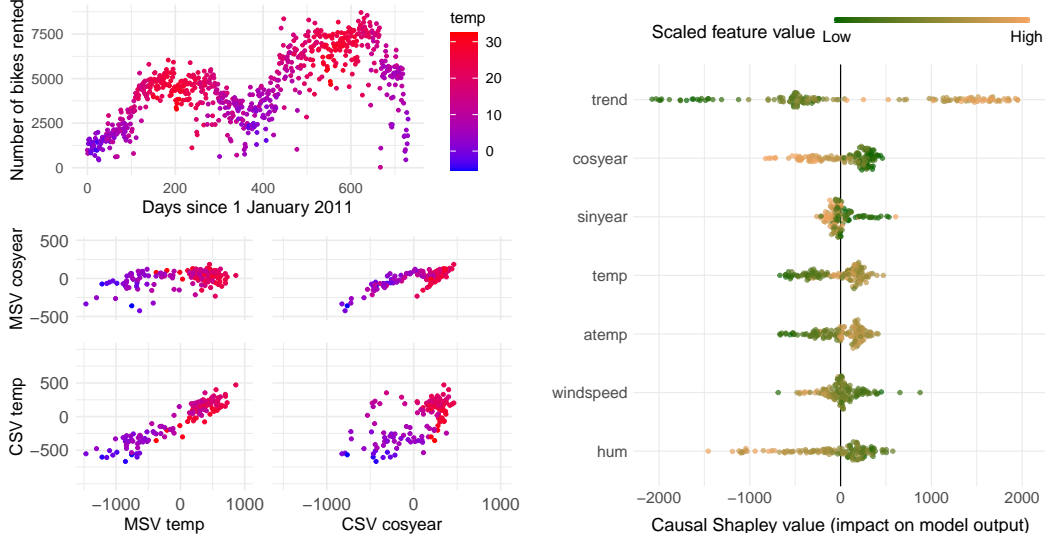

Figure 3: Bike shares in Washington, D.C. in 2011-2012 (top left; colorbar with temperature in degrees Celsius). Sina plot of causal Shapley values for a trained XGBoost model, where the top three date-related variables are considered to be a potential cause of the four weather-related variables (right). Scatter plots of marginal (MSV) versus causal Shapley values (CSV) for temperature (*temp*) and one of the seasonal variables (*cosyear*) show that MSVs almost purely explain the predictions based on temperature, whereas CSVs also give credit to season (bottom left).

where the number above each equal sign refers to the standard *do*-calculus rule from [24] that is applied. For a chain component with dependencies induced by a common confounder, rule (3) applies once more and yields $P(\mathbf{X}_{\tau \cap \bar{S}} | \mathbf{X}_{pa(\tau) \cap \bar{S}}, \mathbf{x}_{pa(\tau) \cap S})$, whereas for a chain component with dependencies induced by mutual interactions, rule (2) again applies and gives $P(\mathbf{X}_{\tau \cap \bar{S}} | \mathbf{X}_{pa(\tau) \cap \bar{S}}, \mathbf{x}_{pa(\tau) \cap S}, \mathbf{x}_{\tau \cap S}))$. □

To compute these observational expectations, we can rely on the various methods that have been proposed to compute conditional Shapley values [1, 6]. Following [1], we will assume a multivariate Gaussian distribution for $P(\mathbf{X})$ that we estimate from the training data. Alternative proposals include assuming a Gaussian copula distribution, estimating from the empirical (conditional) distribution (both from [1]) and a variational autoencoder [6].

## 6 Illustration on real-world data

To illustrate the difference between marginal and causal Shapley values, we consider the bike rental dataset from [5], where we take as features the number of days since January 2011 (*trend*), two cyclical variables to represent season (*cosyear*, *sinyear*), the temperature (*temp*), feeling temperature (*atemp*), wind speed (*windspeed*), and humidity (*hum*). As can be seen from the time series itself (top left plot in Figure 3), the bike rental is strongly seasonal and shows an upward trend. Data was randomly split in 80% training and 20% test set. We trained an XGBoost model for 100 rounds.

We adapted the R package SHAPR from [1] to compute causal Shapley values, which essentially boiled down to an adaptation of the sampling procedure so that it draws samples from the interventional conditional distribution (7) instead of from a conventional observational conditional distribution. The sina plot on the righthand side of Figure 3 shows the causal Shapley values calculated for the trained XGBoost model on the test data. For this simulation, we chose the partial order ({*trend*}, {*cosyear*, *sinyear*}, {all weather variables}), with confounding for the second component and no confounding for the third, to represent that season has an effect on weather, but that we have no clue how to represent the intricate relations between the various weather variables. The sina plot clearly shows the relevance of the trend and the season (in particular cosine of the year, which is -1 on January 1 and +1 on July 1). The scatter plots on the left zoom in on the causal (CSV) and marginal Shapley values (MSV) for *cosyear* and *temp*. The marginal Shapley values for *cosyear* vary

over a much smaller range than the causal Shapley values for *cosyear*, and vice versa for the Shapley values for *temp*: where the marginal Shapley values explain the predictions predominantly based on temperature, the causal Shapley values give season much more credit for the higher bike rental in summer and the lower bike rental in winter. Sina plots for marginal and asymmetric conditional Shapley values can be found in the supplement.

The difference between asymmetric (conditional, from [6]), (symmetric) causal, and marginal Shapley values clearly shows when we consider two days, October 10 and December 3, 2012, with more or less the same temperature of 13 and 13.27 degrees Celsius, and predicted bike counts of 6117 and 6241, respectively. The temperature and predicted bike counts are relatively low for October, yet high for December. The various Shapley values for *cosyear* and *temp* are shown in Figure 4. The marginal Shapley values provide more or less the same explanation for both days, essentially only considering the more direct effect *temp*. The asymmetric conditional Shapley values, which are almost indistinguishable from the asymmetric causal Shapley values in this case, put a huge emphasis on the 'root' cause *cosyear*. The (symmetric) causal Shapley values nicely balance the two extremes, giving credit to both season and temperature, to provide a sensible, but still different explanation for the two days.

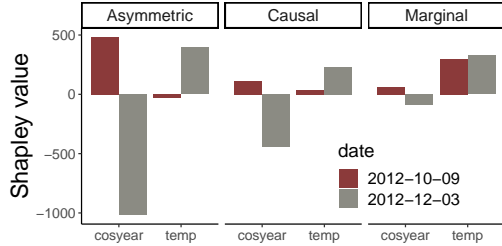

Figure 4: Asymmetric (conditional), (symmetric) causal and marginal Shapley values for two different days, one in October (brown) and one in December (gray) with more or less the same temperature of 13 degrees Celsius. Asymmetric Shapley values focus on the root cause, marginal Shapley values on the more direct effect, and symmetric causal Shapley consider both for the more natural explanation.

## 7    Discussion

In real-world systems, understanding *why* things happen typically implies a causal perspective. It means distinguishing between important, contributing factors and irrelevant side effects. Similarly, understanding why a certain instance leads to a given output by a complex algorithm asks for those features that carry a significant amount of information contributing to the final outcome. Our insight was to recognize the need to properly account for the underlying causal structure between the features in order to derive meaningful and relevant attributive properties in the context of a complex algorithm.

For that, this paper introduced causal Shapley values, a model-agnostic approach to split a model's prediction of the target variable for an individual data point into contributions of the features that are used as input to the model, where each contribution aims to estimate the total effect of that feature on the target and can be decomposed into a direct and an indirect effect. We contrasted causal Shapley values with (interventional interpretations of) marginal and (asymmetric variants of) conditional Shapley values. We proposed a novel algorithm to compute these causal Shapley values, based on causal chain graphs. All that a practitioner needs to provide is a partial causal order (as for asymmetric Shapley values) and a way to interpret dependencies between features that are on an equal footing. Existing code for computing conditional Shapley values is easily generalized to causal Shapley values, without additional computational complexity. Computing conditional and causal Shapley values can be considerably more expensive than computing marginal Shapley values due to the need to sample from conditional instead of marginal distributions, even when integrated with computationally efficient approaches such as KernelSHAP [19] and TreeExplainer [17].

Our approach should be a promising step in providing clear and intuitive explanations for predictions made by a wide variety of complex algorithms, that fits well with natural human understanding and expectations. Additional user studies should confirm to what extent explanations provided by causal Shapley values align with the needs and requirements of practitioners in real-world settings. Similar ideas may also be applied to improve current approaches for (interactive) counterfactual explanations [33] and properly distinguish between direct and total effects of features on a model's prediction. If successful, causal approaches that better match human intuition may help to build much needed trust in the decisions and recommendations of powerful modern-day algorithms.

## Broader Impact

Our research, which aims to provide an explanation for complex machine learning models that can be understood by humans, falls within the scope of explainable AI (XAI). XAI methods like ours can help to open up the infamous "black box" of complicated machine learning models like deep neural networks and decision tree ensembles. A better understanding of the predictions generated by such models may provide higher trust [26], detect flaws and biases [12], higher accuracy [2], and even address the legal "right for an explanation" as formulated in the GDPR [32].

Despite their good intentions, explanation methods do come with associated risks. Almost by definition, any sensible explanation of a complex machine learning system involves some simplification and hence must sacrifice some accuracy. It is important to better understand what these limitations are [11]. Model-agnostic general purpose explanation tools are often applied without properly understanding their limitations and over-trusted [10]. They could possibly even be misused just to check a mark in internal or external audits. Automated explanations can further give an unjust sense of transparency, sometimes referred to as the 'transparency fallacy' [4]: overestimating one's actual understanding of the system. Last but not least, tools for explainable AI are still mostly used as an internal resource by engineers and developers to identify and reconcile errors [2].

Causality is essential to understanding any process and system, including complex machine learning models. Humans have a strong tendency to reason about their environment and to frame explanations in causal terms [28, 16] and causal-model theories fit well to how humans, for example, classify objects [25]. In that sense, explanation approaches like ours, that appeal to a human's capability for causal reasoning should represent a step in the right direction [21].

## Acknowledgments and Disclosure of Funding

This research has been partially financed by the Netherlands Organisation for Scientific Research (NWO), under project 617.001.451.

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
