[Supplementary Material 1]

# Supplement of "Causal Shapley Values: Exploiting Causal Knowledge to Explain Individual Predictions of Complex Models"

**Tom Heskes**
Radboud University
tom.heskes@ru.nl

**Evi Sijben**
Machine2Learn
evisijben@gmail.com

**Ioan Gabriel Bucur**
Radboud University
g.bucur@cs.ru.nl

**Tom Claassen**
Radboud University
t.claassen@science.ru.nl

## 1  *Do*-calculus for cyclic graphs

For completeness, we here repeat the rules of *do*-calculus for cyclic graphs, in the notation of the generalized ID algorithm of [2], which generalizes [5]. We are given a causal graph $G$. To each node $X_i$ which is intervened upon, we add an 'intervention node' $I_{X_i}$, with a directed edge from $I_{X_i}$ to $X_i$ that we clamp to the value $x_i$. The corresponding graph is called $\hat{G}^+$. $\hat{G}_{do(\mathbf{W})}$ is now obtained by removing from $\hat{G}^+$ all incoming edges to variables that are part of $\mathbf{W}$, except those from the corresponding intervention nodes $I_{\mathbf{W}}$. We use shorthand

$$\mathbf{Y} \underset{G}{\overset{\sigma}{\perp\!\!\!\perp}} \mathbf{X} \mid \mathbf{Z}, do(\mathbf{W})$$

to indicate that $\mathbf{Y}$ and $\mathbf{X}$ are $\sigma$-separated by $\mathbf{Z}$ in the graph $\hat{G}_{do(\mathbf{W})}$. $\sigma$-separation is a generalization of standard d-separation (see [2] for details).

*Do*-calculus now consists of the following three inference rules that can be used to map interventional and observational distributions.

1. Insertion/deletion of observation:

$$P(\mathbf{Y}|\mathbf{X}, \mathbf{Z}, do(\mathbf{W})) = P(\mathbf{Y}|\mathbf{Z}, do(\mathbf{W})) \ \text{ if } \ \mathbf{Y} \underset{G}{\overset{\sigma}{\perp\!\!\!\perp}} \mathbf{X} \mid \mathbf{Z}, do(\mathbf{W}) \ .$$

2. Action/observation exchange:

$$P(\mathbf{Y}|do(\mathbf{X}), \mathbf{Z}, do(\mathbf{W})) = P(\mathbf{Y}|\mathbf{X}, \mathbf{Z}, do(\mathbf{W})) \ \text{ if } \ \mathbf{Y} \underset{G}{\overset{\sigma}{\perp\!\!\!\perp}} I_{\mathbf{X}} \mid \mathbf{X}, \mathbf{Z}, do(\mathbf{W}) \ .$$

3. Insertion/deletion of actions:

$$P(\mathbf{Y}|do(\mathbf{X}), \mathbf{Z}, do(\mathbf{W})) = P(\mathbf{Y}|\mathbf{Z}, do(\mathbf{W})) \ \text{ if } \ \mathbf{Y} \underset{G}{\overset{\sigma}{\perp\!\!\!\perp}} I_{\mathbf{X}} \mid \mathbf{Z}, do(\mathbf{W}) \ .$$

Through consecutive application of these rules, we can try to turn any interventional probability of interest into an observational probability.

Figure 1: Three causal models with the same observational distribution over features, yet a different causal structure. To connect to the models in the main text, we set $\beta_1 = 0$ and $\beta_2 = \beta$, except that for the 'fork' we set $\beta_2 = 0, \beta_1 = \beta$, and then swap the indices.

## 2 Shapley values for linear models

We will make use of the *do*-calculus rules above to derive the causal Shapley values for the four different models in Figure 1 in the main text. To this end, we consider the three models in Figure 1 that predict $f(x_1, x_2) = \beta_1 x_1 + \beta_2 x_2$ for general values of $\beta_1$ and $\beta_2$. All three models have the same observational probability distribution, with $\mathbb{E}[X_i] = \bar{x}_i$ and $\mathbb{E}[X_{3-i}|X_i = x_i] = \alpha_i x_i$, for $i = 1, 2$, yet different causal structures. We will arrive at the main text's results for the 'chain', 'confounder', and 'cycle' by setting $\beta_1 = 0$, whereas for the 'fork' we set $\beta_2 = 0$ and swap the two indices. We then further need to take $\bar{x}_1 = \bar{x}_2 = 0$, and $\alpha = \alpha_2$.

Following the definitions in the main text, the contribution of feature $i$ given permutation $\pi$ is the difference in value function before and after setting the feature to its value:

$$\phi_i(\pi) = v(\{j : j \preceq_\pi i\}) - v(\{j : j \prec_\pi i\}),\tag{1}$$

with value function

$$v(S) = \mathbb{E}\left[f(\mathbf{X})|do(\mathbf{X}_S = \mathbf{x}_S)\right] = \int d\mathbf{X}_{\bar{S}}\ P(\mathbf{X}_{\bar{S}}|\hat{\mathbf{x}}_S)f(\mathbf{X}_{\bar{S}}, \mathbf{x}_S),\tag{2}$$

where we use shorthand $\hat{\mathbf{x}}$ for $do(\mathbf{X} = \mathbf{x})$. Combining these two definitions and substituting $f(\mathbf{x}) = \sum_i \beta_i x_i$, we obtain

$$\phi_i(\pi) = \beta_i\left(x_i - \mathbb{E}[X_i|\hat{\mathbf{x}}_{j:j<_\pi i}]\right) + \sum_{k >_\pi i}\beta_k\left(\mathbb{E}[X_k|\hat{\mathbf{x}}_{j:j\leq_\pi i}] - \mathbb{E}[X_k|\hat{\mathbf{x}}_{j:j<_\pi i}]\right).$$

The first term corresponds to the direct effect, the second one to the indirect effect. Symmetric causal Shapley values will follow by averaging these contributions for the two possible permutations $\pi = (1, 2)$ and $\pi = (2, 1)$. Conditional Shapley values result when replacing conditioning by intervention with conventional conditioning by observation, marginal Shapley values by not conditioning at all.

To start with the latter, we immediately see that for *marginal Shapley values* the indirect effect vanishes and the direct effect simplifies to

$$\phi_i = \phi_i(\pi) = \beta_i(x_i - \mathbb{E}[X_i]) = \beta_i(x_i - \bar{x}_i),$$

as also derived in [1].

For symmetric conditional Shapley values, we do get different contributions for the two different permutations, but by definition still the same for the three different models:

$$\phi_1(1, 2) = \beta_1(x_1 - \mathbb{E}[X_1]) + \beta_2(\mathbb{E}[X_2|x_1] - \mathbb{E}[X_2]) = \beta_1(x_1 - \bar{x}_1) + \beta_2\alpha_1(x_1 - \bar{x}_1)$$
$$\phi_2(1, 2) = \beta_2(x_2 - \mathbb{E}[X_2|x_1]) = \beta_2(x_2 - \bar{x}_2) - \beta_2\alpha_1(x_1 - \bar{x}_1).\tag{3}$$

Here the first term in the contribution for the first feature corresponds to the direct effect and the second term to the indirect effect. The contribution for the second feature only consists of a direct

| expectation | chain | confounder | cycle |
|---|---|---|---|
| $\mathbb{E}[X_1|\hat{x}_2]$ | $\mathbb{E}[X_1]$ | $\mathbb{E}[X_1]$ | $\mathbb{E}[X_1|x_2]$ |
| $\mathbb{E}[X_2|\hat{x}_1]$ | $\mathbb{E}[X_2|x_1]$ | $\mathbb{E}[X_2]$ | $\mathbb{E}[X_2|x_1]$ |

Table 1: Turning expectations under conditioning by intervention into expectations under conventional conditioning by observation for the three models in Figure 1.

effect. The contributions for the other permutation follow by swapping the indices and the final Shapley values by averaging to arrive at the *symmetric conditional Shapley values*

$$\phi_1 = \beta_1(x_1 - \bar{x}_1) - \frac{1}{2}\beta_1\alpha_2(x_2 - \bar{x}_2) + \frac{1}{2}\beta_2\alpha_1(x_1 - \bar{x}_1)$$

$$\phi_2 = \beta_2(x_2 - \bar{x}_2) - \frac{1}{2}\beta_2\alpha_1(x_1 - \bar{x}_1) + \frac{1}{2}\beta_1\alpha_2(x_2 - \bar{x}_2) , \tag{4}$$

where now the first two terms constitute the direct effect and the third term the indirect effect.

The *asymmetric conditional Shapley values* consider both permutations for the confounder and the cycle, and hence are equivalent to the symmetric Shapley values for those models. Yet for the chain, they only consider the permutation $\pi(1, 2)$ and thus yield $\phi = \phi(1, 2)$ from (3).

To go from the symmetric conditional Shapley values to the causal symmetric Shapley values, we follow the same line of reasoning, but have to replace $\mathbb{E}[X_2|x_1]$ by $\mathbb{E}[X_2|\hat{x}_1]$ and $\mathbb{E}[X_1|x_2]$ by $\mathbb{E}[X_1|\hat{x}_2]$. Table 1 tells whether the expectations under conditioning by intervention reduce to expectations under conditioning by observation (because of the second rule of *do*-calculus above) or to marginal expectations (because of the third rule). For the chain we have

$$P(X_2|\hat{x}_1) = P(X_2|x_1) \text{ since } X_2 \overset{\sigma}{\underset{G}{\perp\!\!\!\perp}} I_{X_1} \mid X_1 \text{ (rule 2), yet } P(X_1|\hat{x}_2) = P(X_1) \text{ since } X_1 \overset{\sigma}{\underset{G}{\perp\!\!\!\perp}} I_{X_2} \text{ (rule 3),}$$

for the confounder

$$P(X_2|\hat{x}_1) = P(X_2) \text{ since } X_2 \overset{\sigma}{\underset{G}{\perp\!\!\!\perp}} I_{X_1} \text{ and } P(X_1|\hat{x}_2) = P(X_1) \text{ since } X_1 \overset{\sigma}{\underset{G}{\perp\!\!\!\perp}} I_{X_2} \text{ (rule 3),}$$

and for the cycle

$$P(X_2|\hat{x}_1) = P(X_2|x_1) \text{ since } X_2 \overset{\sigma}{\underset{G}{\perp\!\!\!\perp}} I_{X_1} \mid X_1 \text{ and } P(X_1|\hat{x}_2) = P(X_1|x_2) \text{ since } X_1 \overset{\sigma}{\underset{G}{\perp\!\!\!\perp}} I_{X_2} \mid X_2 \text{ (rule 2).}$$

Consequently, for the confounder the *symmetric* and *asymmetric causal Shapley values* coincide with the marginal Shapley values (consistent with [4]) and for the cycle with the symmetric conditional Shapley values from (4). For the chain, the causal symmetric Shapley values become

$$\phi_1(1, 2) = \beta_1(x_1 - \bar{x}_1) + \frac{1}{2}\beta_2\alpha_1(x_1 - \bar{x}_1)$$

$$\phi_2(1, 2) = \beta_2(x_2 - \bar{x}_2) - \frac{1}{2}\beta_2\alpha_1(x_1 - \bar{x}_1) , \tag{5}$$

where the asymmetric causal Shapley values coincides with the asymmetric conditional Shapley values from (5).

Collecting all results and setting $\bar{x}_1 = \bar{x}_2 = \beta_1 = 0, \beta_2 = \beta$, and $\alpha_1 = \alpha$ (after swapping the indices for the 'fork'), we arrive at the Shapley values reported in Figure 1 in the main text. Note that for most Shapley values, the indirect effect for the second feature vanishes because we chose to set $\beta_1 = 0$. The exceptions, apart from the marginal Shapley values, are the causal Shapley values for the chain and the confounder, as well as the asymmetric conditional Shapley values for the chain: these show no indirect effect for the second feature even for nonzero $\beta_1$.

## 3 Proofs and corollaries on causal chain graphs

In this section we expand on the proof of Theorem 1 in the main text and add some corollaries to link back to other approaches for computing Shapley values.

The probability distribution for a causal chain graph boils down to a directed acyclic graph of chain components:

$$P(\mathbf{X}) = \prod_{\tau \in \mathcal{T}} P(\mathbf{X}_\tau|\mathbf{X}_{pa(\tau)}) . \tag{6}$$

For each (fully connected) chain component, we further need to specify whether (surplus) dependencies within the component are due to confounding or due to mutual interactions. Given this information, we can turn any causal query into an observational distribution with the following interventional formula.

**Theorem 1.** *For causal chain graphs, we have the interventional formula*

$$P(\mathbf{X}_{\bar{S}}|do(\mathbf{X}_S = \mathbf{x}_S)) = \prod_{\tau \in \mathcal{T}_{\text{confounding}}} P(\mathbf{X}_{\tau \cap \bar{S}}|\mathbf{X}_{pa(\tau) \cap \bar{S}}, \mathbf{x}_{pa(\tau) \cap S}) \times$$

$$\prod_{\tau \in \mathcal{T}_{\overline{\text{confounding}}}} P(\mathbf{X}_{\tau \cap \bar{S}}|\mathbf{X}_{pa(\tau) \cap \bar{S}}, \mathbf{x}_{pa(\tau) \cap S}, \mathbf{x}_{\tau \cap S}) . \tag{7}$$

*Proof.* Plugging in (6) and using shorthand $\hat{\mathbf{x}} = do(\mathbf{X} = \mathbf{x})$, we obtain

$$P(\mathbf{X}_{\bar{S}}|\hat{\mathbf{x}}_S) = P(\mathbf{X}|\hat{\mathbf{x}}_S) = \prod_{\tau \in \mathcal{T}} P(\mathbf{X}_\tau|\mathbf{X}_{\tau' <_G \tau}, \hat{\mathbf{x}}_S) \stackrel{(1)}{=} \prod_{\tau \in \mathcal{T}} P(\mathbf{X}_\tau|\mathbf{X}_{pa(\tau)}, \hat{\mathbf{x}}_S) = \prod_{\tau \in \mathcal{T}} P(\mathbf{X}_{\tau \cap \bar{S}}|\mathbf{X}_{pa(\tau) \cap \bar{S}}, \hat{\mathbf{x}}_S) ,$$

where in the second step we made use of *do*-calculus rule (1): the conditional independencies in the causal chain graph $G$ are preserved when we intervene on some of the variables. Now rule (3) tells us that we can ignore any interventions from nodes in components further down the causal chain graph as well as those from higher up that are shielded by the direct parents:

$$P(\mathbf{X}_{\tau \cap \bar{S}}|\mathbf{X}_{pa(\tau) \cap \bar{S}}, \hat{\mathbf{x}}_S) \stackrel{(3)}{=} P(\mathbf{X}_{\tau \cap \bar{S}}|\mathbf{X}_{pa(\tau) \cap \bar{S}}, \hat{\mathbf{x}}_{pa(\tau) \cap S}, \hat{\mathbf{x}}_{\tau \cap S}) .$$

Rule (2) then states that conditioning by intervention upon variables higher up in the causal chain graph is equivalent to conditioning by observation:

$$P(\mathbf{X}_{\tau \cap \bar{S}}|\mathbf{X}_{pa(\tau) \cap \bar{S}}, \hat{\mathbf{x}}_{pa(\tau) \cap S}, \hat{\mathbf{x}}_{\tau \cap S}) \stackrel{(2)}{=} P(\mathbf{X}_{\tau \cap \bar{S}}|\mathbf{X}_{pa(\tau) \cap \bar{S}}, \mathbf{x}_{pa(\tau) \cap S}, \hat{\mathbf{x}}_{\tau \cap S}) .$$

For a chain component with dependencies induced by a common confounder, rule (3) applies once more and makes that we can ignore the interventions:

$$P(\mathbf{X}_{\tau \cap \bar{S}}|\mathbf{X}_{pa(\tau) \cap \bar{S}}, \mathbf{x}_{pa(\tau) \cap S}, \hat{\mathbf{x}}_{\tau \cap S}) = P(\mathbf{X}_{\tau \cap \bar{S}}|\mathbf{X}_{pa(\tau) \cap \bar{S}}, \mathbf{x}_{pa(\tau) \cap S}) .$$

For a chain component with dependencies induced by mutual interactions, rule (2) again applies and allows us to replace conditioning by intervention with conditioning by observation:

$$P(\mathbf{X}_{\tau \cap \bar{S}}|\mathbf{X}_{pa(\tau) \cap \bar{S}}, \mathbf{x}_{pa(\tau) \cap S}, \hat{\mathbf{x}}_{\tau \cap S}) = P(\mathbf{X}_{\tau \cap \bar{S}}|\mathbf{X}_{pa(\tau) \cap \bar{S}}, \mathbf{x}_{pa(\tau) \cap S}, \mathbf{x}_{\tau \cap S})) .$$

$\square$

Algorithm 1 provides pseudocode on how to estimate the value function $v(S)$ by drawing samples from the interventional probability (7). It assumes that a user has specified a partial causal ordering of the features, which is translated to a chain graph with components $\mathcal{T}$, and for each (non-singleton) component $\tau$ whether or not surplus dependencies are the result of confounding. Other prerequisites include access to the model $f()$, the feature vector $\mathbf{x}$, (a procedure to sample from) the observational probability distribution $P(\mathbf{X})$, and the number of samples $N_{\text{samples}}$.

Theorem 1 connects to observations made and algorithms proposed in recent papers.

**Corollary 1.** *With all features combined in a single component and all dependencies induced by confounding, as in [4], causal Shapley values are equivalent to marginal Shapley values.*

*Proof.* With just a single confounded component $\tau$, $pa(\tau) = \emptyset$ and (7) reduces to $P(\mathbf{X}_{\bar{S}})$. $\square$

**Corollary 2.** *With all features combined in a single component and all dependencies induced by mutual interactions, causal Shapley values are equivalent to conditional Shapley values as proposed in [1].*

*Proof.* With just a single non-confounded component $\tau$, $pa(\tau) = \emptyset$ and (7) reduces to $P(\mathbf{X}_{\bar{S}}|\mathbf{x}_S)$. $\square$

**Corollary 3.** *When we only take into account permutations that match the causal ordering and assume that all dependencies within chain components are induced by mutual interactions, the resulting asymmetric causal Shapley values are equivalent to the asymmetric conditional Shapley values as defined in [3].*

**Algorithm 1** Compute the value function $v(S)$ under conditioning by intervention.

1: **function** VALUEFUNCTION($S$)
2:     **for** $k \leftarrow 1$ to $N_{\text{samples}}$ **do**
3:         **for all** $j \leftarrow 1$ to $|\mathcal{T}|$ **do**                    ▷ run over all chain components in their causal order
4:             **if** confounding($\tau_j$) **then**
5:                 **for all** $i \in \tau_j \cap \bar{S}$ **do**
6:                     Sample $\tilde{x}_i^{(k)} \sim P(X_i | \tilde{\mathbf{x}}_{pa(\tau_j) \cap \bar{S}}^{(k)}, \mathbf{x}_{pa(\tau_j) \cap S})$        ▷ can be drawn independently
7:                 **end for**
8:             **else**
9:                 Sample $\tilde{\mathbf{x}}_{\tau_j \cap \bar{S}}^{(k)} \sim P(\mathbf{X}_{\tau_j \cap \bar{S}} | \tilde{\mathbf{x}}_{pa(\tau_j) \cap \bar{S}}^{(k)}, \mathbf{x}_{pa(\tau_j) \cap S}, \mathbf{x}_{\tau_j \cap S})$        ▷ e.g., Gibbs sampling
10:             **end if**
11:         **end for**
12:     **end for**
13:     $v \leftarrow \dfrac{1}{N_{\text{samples}}} \displaystyle\sum_{k=1}^{N_{\text{samples}}} f(\mathbf{x}_S, \tilde{\mathbf{x}}_{\bar{S}}^{(k)})$
14:     **return** $v$
15: **end function**

*Proof.* Following [3], asymmetric Shapley values only include those permutations $\pi$ for which $i \prec_\pi j$ for all known ancestors $i$ of descendants $j$ in the causal graph. For those permutations, we are guaranteed to have $\tau \prec_G \tau'$ for all $\tau, \tau' \in \mathcal{T}$ such that $\tau \cap S \neq \emptyset, \tau' \cap \bar{S} \neq \emptyset$. That is, the chain components that contain features from $S$ are never later in the causal ordering of the chain graph $G$ than those that contain features from $\bar{S}$. We then have

$$P(\mathbf{X}_{\bar{S}} | \mathbf{x}_S) = \prod_{\tau \in \mathcal{T}} P(\mathbf{X}_{\tau \cap \bar{S}} | \mathbf{X}_{pa(\tau) \cap \bar{S}}, \mathbf{x}_S) = \prod_{\tau \in \mathcal{T}} P(\mathbf{X}_{\tau \cap \bar{S}} | \mathbf{X}_{pa(\tau) \cap \bar{S}}, \mathbf{x}_{pa(\tau) \cap S}, \mathbf{x}_{\tau \cap S}) = P(\mathbf{X}_{\bar{S}} | \hat{\mathbf{x}}_S),$$

where in the last step we used interventional formula (7) in combination with the fact that $\mathcal{T}_{\text{confounding}} = \emptyset$.                                                                                                           $\square$

## 4   Additional illustrations on the bike rental data

Figure 2 shows sina plots for asymmetric conditional Shapley values (left) and marginal Shapley values (right), to be compared with the sina plots for symmetric causal Shapley values in Figure 3 of the main text. In this case, the sina plots for asymmetric causal Shapley values are virtually indistinguishable from those for the asymmetric conditional Shapley values.

It can be seen that the marginal Shapley values strongly focus on temperature, largely ignoring the seasonal variables. The asymmetric Shapley values do the opposite: they focus on the seasonal variables, in particular *cosyear* and put much less emphasis on the temperature variables.

## 5   Comparing symmetric and asymmetric Shapley values on the XOR problem

We consider the standard XOR problem with binary features $X_1$ and $X_2$ and binary output $Y$:

| $X_1$ | $X_2$ | Y |
|-------|-------|---|
| 0 | 0 | 0 |
| 0 | 1 | 1 |
| 1 | 0 | 1 |
| 1 | 1 | 0 |

We generate a dataset of $n$ samples by drawing features and corresponding outputs with probabilities $p_{ij} = P(X_1 = i, X_2 = j)$. We will choose $p_{00} = p_{11} = \frac{1}{4}(1 + \epsilon)$ and $p_{01} = p_{10} = \frac{1}{4}(1 - \epsilon)$. With $\epsilon > 0$, the probability of the two features having the same values is larger than the probability of them having different values. $\hat{p}_{ij}$ is the same probability estimated from the data, e.g., by computing the

Figure 2: Sina plots of asymmetric (conditional) Shapley values (left) and marginal Shapley values (right). See Figure 3 in the main text for further details.

frequencies of the four input combinations. We train a neural network on the generated data, which yields a function $\hat{f}(X_1, X_2)$ hopefully closely approximating the correct XOR function. The parameter $\epsilon$ captures the dependency between the two features and can be interpreted as a measure of the causal strength when the two features are causally related.

We will now compute the various Shapley values for the data point $(X_1, X_2) = (0, 0)$. The value functions with all features either 'out-of-coalition' or 'in-coalition' are the same for all Shapley values:

$$v(\{\}) \;=\; \mathbb{E}\left[f(\mathbf{X})\right] = \sum_{i,j} \hat{p}_{ij}\hat{f}(i, j) \approx \frac{1}{2}(1 - \epsilon)$$

$$v(\{1, 2\}) \;=\; \hat{f}(0, 0) \approx 0 \,,$$

where we use the convention that the Shapley values computed from the fitted probabilities and learned neural network appear before the $\approx$-sign, and those that we obtain when the fitted probabilities equal the probabilities used to generate the data and when the learned neural network equals the XOR function after the $\approx$-sign.

The value functions for the case that one input is 'in-coalition' and the other 'out-of-coalition' does depend on the type of Shapley value under consideration. For the marginal Shapley values we get

$$v(\{1\}) \;=\; \mathbb{E}\left[f(0, X_2)\right] = \sum_{j}\left(\sum_{i} \hat{p}_{ij}\right)\hat{f}(0, j) \approx \frac{1}{2}$$

$$v(\{2\}) \;=\; \mathbb{E}\left[f(X_1, 0)\right] = \sum_{i}\left(\sum_{j} \hat{p}_{ij}\right)\hat{f}(i, 0) \approx \frac{1}{2} \,, \tag{8}$$

yet for the conditional Shapley values

$$v(\{1\}) \;=\; \mathbb{E}\left[f(0, X_2)|X_1 = 0\right] = \sum_{j} \frac{\hat{p}_{0j}}{\sum_{i} \hat{p}_{ij}}\hat{f}(0, j) \approx \frac{1}{2}(1 - \epsilon)$$

$$v(\{2\}) \;=\; \mathbb{E}\left[f(X_1, 0)|X_2 = 0\right] = \sum_{i} \frac{\hat{p}_{i0}}{\sum_{j} \hat{p}_{ij}}\hat{f}(i, 0) \approx \frac{1}{2}(1 - \epsilon) \,. \tag{9}$$

The value functions for the causal Shapley values depend on the presumed causal model that generates the dependencies. In case the dependencies are assumed to be the result of confounding, we get the value functions in (8) as for the marginal Shapley values and when the dependencies are assumed to

Figure 3: The conditional symmetric, causal symmetric and (causal/conditional) asymmetric Shapley values of data point $(X_1, X_2) = (0, 0)$ under the assumption of a causal chain $X_1 \rightarrow X_2$ for different causal strengths $\epsilon$. The bars indicate the mean Shapley value and standard deviation of 100 runs with a neural network trained on 100 data points generated according to a particular causal strength $\epsilon$. Red lines give the theoretical Shapley values, to be expected for an infinite amount of samples and when the neural networks would have perfectly learned to represent the XOR function. The third row gives the difference between the Shapley values for $X_1$ and $X_2$ and clearly shows the discontinuity of asymmetric Shapley values for $\epsilon = 0$.

be the result of mutual interaction the value functions in (9) as for the conditional Shapley values. The more interesting case is when we assume a causal chain, e.g., $X_1 \rightarrow X_2$:

$$v(\{1\}) \quad = \quad \mathbb{E}\left[f(0, X_2)|do(X_1 = 0)\right] = \mathbb{E}\left[f(0, X_2)|X_1 = 0\right] = \sum_j \frac{\hat{p}_{0j}}{\sum_i \hat{p}_{ij}} \hat{f}(0, j) \approx \frac{1}{2}(1 - \epsilon)$$

$$v(\{2\}) \quad = \quad \mathbb{E}\left[f(X_1, 0)|do(X_2 = 0)\right] = \mathbb{E}\left[f(X_1, 0)\right] = \sum_i \left(\sum_j \hat{p}_{ij}\right) \hat{f}(i, 0) \approx \frac{1}{2}, \qquad (10)$$

and the same with indices 1 and 2 interchanged for the causal chain $X_2 \rightarrow X_1$.

Given these value functions, we can now compute the various Shapley values. For marginal and symmetric Shapley values we have

$$\phi_1 \quad = \quad \frac{1}{2}[v(\{1\}) - v(\{\})] + \frac{1}{2}[v(\{1, 2\}) - v(\{2\})]$$

$$\phi_2 \quad = \quad \frac{1}{2}[v(\{2\}) - v(\{\})] + \frac{1}{2}[v(\{1, 2\}) - v(\{1\}]) ,$$

whereas for asymmetric Shapley values, assuming the causal chain $X_1 \rightarrow X_2$,

$$\phi_1 \quad = \quad v(\{1\}) - v(\{\})$$

$$\phi_2 \quad = \quad v(\{1, 2\}) - v(\{1\}) ,$$

and the same with indices 1 and 2 interchanged for the causal chain $X_2 \rightarrow X_1$.

With the expressions above, we can compute the various Shapley values based on a learned neural network and the actual frequencies of the generated feature combinations and compare those with the theoretical values obtained when the estimated frequencies equal the probabilities used to generate the

data and the neural network indeed managed to learn the XOR function. For the latter we distinguish the following cases.

**identical:** $\phi_1 = \phi_2 \approx \frac{1}{4}\epsilon - \frac{1}{4}$. This applies to marginal, symmetric conditional, symmetric causal assuming confounding, symmetric causal assuming mutual interaction.

**symmetric causal:** $\phi_1 \approx -\frac{1}{4}$ and $\phi_2 \approx \frac{1}{2}\epsilon - \frac{1}{4}$ assuming the causal chain $X_1 \to X_2$ and vice versa for $X_1 \to X_2$.

**asymmetric:** $\phi_1 \approx 0$ and $\phi_2 \approx \frac{1}{2}\epsilon - \frac{1}{2}$ assuming the causal chain $X_1 \to X_2$ and vice versa for $X_1 \to X_2$. These apply both to asymmetric conditional and asymmetric causal.

In this example, symmetric causal Shapley values are clearly to be preferred over asymmetric causal Shapley values for small causal strengths. Inserting a causal link with zero strength ($\epsilon = 0$), asymmetric Shapley values jump from the symmetric $\phi_1 = \phi_2 \approx -\frac{1}{4}$ to the completely asymmetric $\phi_1 \approx 0$ and $\phi_2 \approx -\frac{1}{2}$, assigning all credit to the second feature, even though the first feature in reality does not affect the second feature at all. Symmetric Shapley values, on the other hand, are insensitive to the insertion of a causal link with zero strength: in the limit $\epsilon \to 0$ symmetric causal Shapley values correctly converge to marginal Shapley values.

Figure 3 shows the results of a series of simulations, computing different Shapley values for trained neural networks and comparing these to the theoretical values. The discontinuity of asymmetric Shapley values (conditional and causal asymmetric Shapley values are identical in this example) is most clearly seen in the third row, showing the difference between the Shapley values for $X_1$ and $X_2$. Symmetric conditional Shapley values do not distinguish between the Shapley values for $X_1$ and $X_2$ for any causal strength $\epsilon$, whereas the symmetric causal Shapley values are identical for $\epsilon = 0$ and then slowly start to deviate for larger values of $\epsilon$.

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

[Supplementary Material 2 · paper.pdf]


![JoSS logo] The Journal of Open Source Software

# shapr: An R-package for explaining machine learning models with dependence-aware Shapley values

**Nikolai Sellereite**[1] **and Martin Jullum**[1]

**1** Norwegian Computing Center

**DOI:** XX.XXXXX/joss.XXXXX

**Software**

- Review ↗
- Repository ↗
- Archive ↗

**Submitted:**
**Published:**

## Summary

A common task within machine learning is to train a model to predict an unknown outcome (response variable) based on a set of known input variables/features. When using such models for real life applications, it is often crucial to understand why a certain set of features lead to a specific prediction. Most machine learning models are, however, complicated and hard to understand, so that they are often viewed as "black-boxes", that produce some output from some input.

Shapley values (Shapley, 1953) is a concept from cooperative game theory used to distribute fairly a joint payoff among the cooperating players. Štrumbelj & Kononenko (2010) and later Lundberg & Lee (2017) proposed to use the Shapley value framework to explain predictions by distributing the prediction value on the input features. Established methods and implementations for explaining predictions with Shapley values like Shapley Sampling Values (Štrumbelj & Kononenko, 2014), SHAP/Kernel SHAP (Lundberg & Lee, 2017), and to some extent TreeSHAP/TreeExplainer (Lundberg et al., 2020; Lundberg, Erion, & Lee, 2018), assume that the features are independent when approximating the Shapley values. The R-package shapr, however, implements the methodology proposed by Aas, Jullum, & Løland (2019), where predictions are explained while accounting for the dependence between the features, resulting in significantly more accurate approximations to the Shapley values.

## Implementation

shapr implements a variant of the Kernel SHAP methodology (Lundberg & Lee, 2017) for efficiently dealing with the combinatorial problem related to the Shapley value formula. The main methodological contribution of Aas et al. (2019) is three different methods to estimate certain conditional expectation quantities, referred to as the *empirical*, *Gaussian* and *copula* approach. Additionally, the user has the option of combining the three approaches. The implementation supports explanation of models fitted with the following functions natively: `stats::lm` (R Core Team, 2019), `stats::glm` (R Core Team, 2019), `ranger::ranger` (Wright & Ziegler, 2017), `mgcv::gam` (Wood, 2017) and `xgboost::xgboost`/`xgboost::xgb.train` (Chen et al., 2019). Moreover, the package supports explanation of custom models by supplying two simple additional class functions.

For reference, the package also includes a benchmark implementation of the original (independence assuming) version of Kernel SHAP (Lundberg & Lee, 2017), providing identical results to the "official" Kernel SHAP `Python` package `shap`. This allows the user to easily see the effect and importance of accounting for the feature dependence.

The user interface in the package has largely been adopted from the R-package `lime` (Pedersen & Benesty, 2019). The user first sets up the explainability framework with the `shapr` function. Then the output from `shapr` is provided to the `explain` function, along with the data to explain the prediction and the method that should be used to estimate the aforementioned conditional expectations.

The majority of the code is in plain `R` (R Core Team, 2019), while the most time consuming operations are coded in `C++` through the `Rcpp` package (Eddelbuettel & François, 2011) and `RcppArmadillo` package (Eddelbuettel & Sanderson, 2014) for computational speed up. For RAM efficiency and computational speed up of typical bookeeping operations, we utilize the `data.table` package (Dowle & Srinivasan, 2019) which does operations "by reference", i.e. without memory copies.

For a detailed description of the underlying methodology that the package implements, we refer to the paper (Aas et al., 2019) which uses the package in examples and simulation studies. To get started with the package, we recommend going through the package vignette and introductory examples available at the package's pkgdown site.

# Acknowledgement

This work was supported by the Norwegian Research Council grant 237718 (Big Insight).