[Reviews · NeurIPS 2020]

Review 1

Summary and Contributions: The paper introduces a new way of calculating Shapley values for local model explanations. The authors propose "causal Shapley values," which differ from existing Shapley value methods in some respects. If I am correct in my understanding of the paper, the paper proposes using a different cooperative game for representing the model's dependence on subsets of features.

Strengths: The paper presents a different perspective on how to integrate causal knowledge into Shapley value-based model explanations. The authors contrast their approach with another recent paper that takes a different approach to integrating causal knowledge, asymmetric Shapley values.

Weaknesses: I re-read this paper several times and found it very hard to follow. In particular, I could not find a part of the paper that clearly defined "causal Shapley values," which makes evaluating the paper rather difficult. I'm open to changing my opinion if the other reviewers disagree, but my impression is that this paper's clarity is a serious problem. I'll elaborate on this and a couple other thoughts below. On the paper's clarity problems: - The paper should be much more clear when describing its proposal, "causal Shapley values." Section 2 seemed to be the place where they were defined, and indeed Section 2 compares them with asymmetric Shapley values (so we should presumably know what they are by this point, right?). But an application of the Shapley value to Eq. 3 is precisely the marginal Shapley value, also known as the interventional Shapley value. So ultimately I couldn't tell whether this section introduces causal Shapley values or not. - Section 2 says "interventional Shapley values conveniently simplify to marginal Shapley values." The discussion about the difference between the interventional approach and the marginal approach is a bit odd. It seems obvious, based on the definition of the do operator, that "conditioning by intervention" is equivalent to using the marginal distribution. Why are these initially presented as two different options? And why are distinctions drawn with Janzing et al. (lines 105-107), given that this approach does precisely the same thing? - Other parts of the paper seem to clearly indicate that causal Shapley values are not an application of the Shapley value to Eq. 3. For example, Section 4 distinguishes between symmetric causal Shapley values and a variety of other Shapley variants. So we should certainly know causal Shapley values are by now, right? - Section 6 says that the Shapley value estimation algorithms were modified to use the conditional interventional distribution, as given by Eq. 7. But the paper has no equation 7! - In the end, my best guess was that causal Shapley values were somehow defined in Section 5. If so, the authors should have been much more clear about this, and probably should not have contrasted causal Shapley values with other variants throughout the paper. The authors might consider explicitly saying something along the lines of "we define causal Shapley values as follows," or "we define causal Shapley values as the Shapley values of the following cooperative game." About the distinction between direct and indirect effects: - This section (Section 3) requires substantial clarification. The extra term that's introduced to distinguish between direct and indirect effects should be explained in more detail. If I'm not mistaken, it seems to be equal to zero. The text below the math suggests that marginal Shapley values, which are equivalent to what has been decomposed here (the interventional Shapley value), has no indirect effect. Does that mean that the two terms on the second line are equal? - Since this direct/indirect effect decomposition is hard to follow, the authors might consider writing the direct/indirect effect decomposition for conditional Shapley values as well, or at the very least putting it in the supplement, rather than just mentioning that such a decomposition can be made. - If the conditional Shapley values permit this decomposition, why is an alternative preferred over them? Is there something wrong with how conditional Shapley values calculate direct and indirect effects? - Overall, I could not tell what this section was trying to show. Was it just trying to show that marginal Shapley values have no indirect effect? Are the authors even talking about causal Shapley values here? About the use of variable orderings in Section 5: using the terminology of causal chain graphs is new relative to the ASV paper, but the outcome of being able to being able to use partial causal orderings is not. ASVs use a very similar partial ordering concept in their experiments. I did not find the empirical comparison of causal Shapley values with other Shapley variants (Figure 4) very convincing. This plot shows just two predictions with just two features' attributions, so this does not provide strong evidence in favor of causal Shapley values.

Correctness: The claims seem correct.

Clarity: The paper is lacking in clarity. After re-reading it several times, I could not find the part of the paper that clearly defines their proposal, causal Shapley values. I'm open to re-evaluating the ideas in this paper if the authors can clarify their contribution, or if the other reviewers found the paper more comprehensible.

Relation to Prior Work: The authors do a good job of citing plenty of relevant work.

Reproducibility: Yes

Additional Feedback: Update: Thanks to the authors for their response and for offering to clarify the introduction of the proposed method. Clarifying Section 2 (particularly the differences with [9]) will significantly improve the readability of the paper. Now that I have a better understanding of what the authors are proposing, my main issue with this work is that it's not clear why this is the right way to incorporate causal information. The current version does not sufficiently justify Causal Shapley values, and it does not make a strong case that Causal Shapley Values provide a better approach than ASVs (either through the theoretical discussion or through the experiments). For example, I was left wondering about the following question. Existing variants of SHAP hold out features so we can simulate how the model behaves without them; why does it make sense to consider *intervening* on the held-in features? I am not asserting that there is no good answer to this question, but the paper seems to assume without justification that this approach is reasonable. I'm keeping my current score because of the combination of limited novelty (the only modification relative to SHAP or ASVs is the underlying cooperative game), the insufficient experiments, but primarily the fact that the approach is not sufficiently justified.


Review 2

Summary and Contributions: The paper describes a way to define causal feature relevance analysis. In contrast to the popular approach with Shapley values it averages only over orderings that are consistent with the causal order and uses Pearl's do-operator to obtain interventional instead of observational probabilities.

Strengths: The paper discusses issues with previous approaches convincingly. The suggested approach is at least worth to be discussed, although it may not be the final word on this complex topic (there probably isn't a final word). The decomposition of total effects into direct and indirect effect is inspiring, also the discussion of chain graphs with confounding and directional edges.

Weaknesses: The discussion of the decomposition of total into direct and indirect effect is interesting, but lacks systematic consideration since it is not obvious how to decompose the total influence into different paths for a larger number of paths. Given my comments below, the paper could be a bit more critical about its own concepts since quantifying influence is a difficult topic and every definition comes with its now weaknesses.

Correctness: As far as I can see the claims are correct.

Clarity: The paper is mostly well-written. I liked particularly the interesting references to the psychological literature. Section 4 is unfortunately the weakest part in this regard, hard to say what its message is supposed to be. I found its last paragraph entirely confusing: why does sampling counterfactuals correspond to *averaging* over ordering in feature attribution quantification?

Relation to Prior Work: ok

Reproducibility: Yes

Additional Feedback: Every quantification of causal influences raises tons of questions. Here are some thoughts on the present one, which I would appreciate to be mentioned: - although averaging over all orderings consistent with the DAG sounds natural, it comes with a serious conceptual issue, namely the discontinuity with respect to inserting arrows of zero strength. Assume X and Y are independent causes of Z. Then there are two possible orderings. Insert an arrow from X to Y with negligible impact, then there is only one ordering. Compare supplement S3 of 'Quantifying causal influences' by Janzing et al. for a similar problem. I propose to comment on this issue for asymmetric SV and mention that symmetric SV do not suffer from this problem. - issues with discontinuity at arrows with zero strength can be avoided by defining strength via the influence of the noise terms, see e.g. https://arxiv.org/abs/2007.00714 However, this solution comes with another conceptual problem: influence quantification then depends on the functional causal model and is no longer uniquely determined by interventional probabilities. Regarding the criticism of ref [9]: it seems to me that it comes with a different scope, namely feature relevance quantification in the sense of understanding an algorithms rather than understanding causal relations in the real world. Maybe one could mention this difference. Note also that it appears at AISTATS, it's already online. One clarification in this regard: I don't think it is helpful to distinguish between 'causal' versus 'interventional' Shapley Values. Here is where the confusion comes from. The conclusion of [9] that interventional conditionals result in marginalisation only applies to that particular context of causally explaining an algorithm. And it only applies there because [9] formally distinguishes between features in the real world and the corresponding inputs of the algorithm. Deviating from this scenario, interventional probabilities don't necessarily result in marginal probabilities since one feature could influence the target of interest indirectly via another one, as the authors correctly point out. - I was struggling with the remark 'not every causal query need be identifiable (see e.g., [24])' if the DAG is known and all nodes are observed, which type of causal queries are not identifiable? I would like to say, however, that generalization chain graphs would not have been the most important point to discuss until the application to DAGs is fully understood. E.g. some conceptual discussion of toy cases that argue for symmetric versus asymmetric Shapley values? But this is more a matter of taste, I also liked the remarks on the chain graph. The authors have addressed most of my concerns. I am a bit struggling to raise my score even further because I have to agree with some concerns regarding the writing raised by the other reviewers. The paper doesn't seem optimally structured, wording doesn't sound optimal (e.g. see my remark on artificially introducing a difference between interventional and causal conditionals). 'Causal interpretation of Shapley Values" sounds strange: why is this a different interpretation of Shapley Values? Why are chain graphs introduced as part of 'a practical example'? Isn't this mainly a generalization which defines a theoretical contribution?


Review 3

Summary and Contributions: The paper considers locally explaining predictions of a predictive model when the features are not independent. The paper proposes a method by which to compute Shapley values when we intervene on each feature, instead of conventional conditioning. The proposed approach considers access to the causal chain graph of the problem at hand, and derives Shapley values of the (intervened) features according to that. The proposed approach is tested on one dataset and one predictive model.

Strengths: The problem and the approach are interesting.

Weaknesses: The theoretical and practical contributions of the paper are very limited.

Correctness: Yes, but only in limited settings.

Clarity: Yes.

Relation to Prior Work: Yes.

Reproducibility: Yes

Additional Feedback: Update after reading the author response: The authors only partially addressed my concerns and did not answer my main points. One of my main concerns was the lack of enough experimentation and I think running a proper set of them would not fit in the revision and requires another round of work. My rating of the paper remains the same. -------------------------------------------------------------------------------------------------------------------- Overall: The paper considers locally explaining predictions of a predictive model when the features are not independent. The paper proposes a method by which to compute Shapley values when we intervene on each feature, instead of conventional conditioning. The proposed approach considers access to the causal chain graph of the problem at hand, and derives Shapley values of the (intervened) features according to that. The proposed approach is tested on one dataset. The problem and the approach are interesting, but the theoretical and practical contributions of the paper are limited. The theoretical contribution is limited to a special case, which limits applicability of the proposed approach. Further, the approach is only tested on one dataset with 7 features and only one trained ML model, i.e., XGBoost. This is not convincing enough. More methodologies, datasets, and a wider application setting must be considered. My assessment of this paper is a clear cut reject. Major: - The proposed methodology depends on cases where we can intervene on variables (do(x); Pearl’s second rung on the ladder of causality; refer to the Book of Why). However, oftentimes, when one needs to provide an explanation for a predictive model, experimentation (do(x)) is not a viable option. One way of dealing with this is using counterfactual inference (Pearl’s rung 3 in the ladder of causation). The proposed methodology would be much more convincing it if in addition to interventions, it would account for counterfactual inference. The authors could show this in their setting by answering the question: What would the predicted bike count be, if the season was spring, and the temperature was 20? - The main contribution of the paper is summarized in Theorem 1. This theorem only holds for causal chain graphs, which are a special case, and make the contribution very limited. How would the proposed methodology work in other situations, such as when there are forks, (un)shielded colliders, etc.? Identifying the admissible set that would make the causal effect identifiable in these cases have been studied in the literature, but are not discussed in this paper. How would the proposed approach deal with such cases? The domain of applicability of the proposed methodology is very limited. I appreciate the discussion of the chain, fork, confounder, and cycle options in Figure 1, but not all of them are not realistic in the case of the provided running example of the paper, i.e., bike rentals. How can temperature causally affect season in the fork? Perhaps, a better example could help. The proposed approach is only tested on one real-world toy data with only 7 features according to the text. This does not show the scalability of the proposed method in practical examples. It is okay to test the approach on an XGBoost model, but we should also see the results on at least one deep neural network too, because neural networks are a main concern among the XAI community. - Causal relationships are asymmetric in nature. For example, if A causes B, then B might not necessarily cause A. It’s not clear why the authors are calling causal values as symmetric. Results of experiments in figure 4 are not convincing. If the temperature for Oct and Dec, in this example, are roughly the same, then the different in predicted bike count should be attributed to, perhaps, the seasonal difference, which is represented by cosine of year in this case. But the causal Shapley values have computed a lower effect (in magnitude) for cosine of year. Why is it the case? And why should we trust the explanation generated by the causal Shapley value instead of Asymmetric Shapley values. Minor: - There is inconsistency in references style. While many journal/conference names follow a capitalized first letter paradigm, such as Advances in Neural Information Processing Systems (reference 20), others do not follow this, such as Knowledge and information systems (reference 31). This inconsistency is apparent in many references. Careless writing, please fix. - Line 230, clicking on “interventional conditional distribution (7)” takes me to Formula number 6, not 7.


Review 4

Summary and Contributions: This paper proposes a causal approach to Shapley values, which are values that are used to explain what features of the input data to a model contributed to the model's output. By using a causal approach, dependencies between features in the input data can be dealt with, which was not possible before.

Strengths: This is a truly great paper. The contribution of this paper is outstanding, both from a conceptual point of view (letting explainable AI benefit from a causality-based conceptualization; I ) and from the results (the computations are feasible and the provided real-world example is sensible). It is a step in the right direction and it is a noticeable step, not just an idea. Secondly, this paper is incredibly well written. It was a delight to read it and I learned a lot in a short amount of time. This is the best paper I've read in a while. Reading it made up for the horrible paper I had to review earlier today.

Weaknesses: Of course one can always request more (more analyses, more datasets, more causal structures to test the approach on) but in my view, this paper is well-rounded and I didn't miss a single thing.

Correctness: I did not spot any mistakes. I cannot vouch for the correctness of the most technical aspects of this work though, as I'm not an expert in Pearl's do-calculus or Shapley values.

Clarity: Thank you for writing such a clear paper.

Relation to Prior Work: It is discussed at length

Reproducibility: Yes

Additional Feedback: Suggestion for improvement: In Fig 3, bottom-left corner plots, make the grid lines are the same for the x- and y-axis, and perhaps change the x-axis tick labels to -500 and 500, too. That would make it easier to see how the range of the values differ (as emphasized in the main text). In the current form, I had to stare at it for a bit to convince myself that the scales are the same and it is not an effect of stretching the x-axis. I'm excited to see in future work, how the comparison to human subjects' judgments will work out.

[Author Response · NeurIPS 2020]

We would like to thank the reviewers for their comments and feedback. We are aware that in a largely conceptual paper
like ours there are subtleties, and highly appreciate the time and effort that the reviewers are putting in to digest these.

**Reviewer #1**: Causal Shapley values (SVs) are defined in Section 2. These do *not* coincide with what [9] and others
call the interventional SVs (marginal SVs in our terminology). Janzing et al. [9] write down the same equation, but
then choose to ignore any dependencies between the features in the real world (e.g., that in summer it tends to be
warmer than in winter). We do choose to incorporate these dependencies and hence cannot simplify to $P(\mathbf{X}_{\bar{S}}|do(\mathbf{X}_S =$
$\mathbf{x}_S)) = P(\mathbf{X}_{\bar{S}})$, but keep $P(\mathbf{X}_{\bar{S}}|do(\mathbf{X}_S = \mathbf{x}_S))$ in our definition of the causal SVs. We will follow the reviewer's
suggestion to make this more explicit in Section 2. This distinction then hopefully also resolves the reviewer's issue
about the indirect effect: it indeed vanishes for marginal SVs, but need not vanish for causal and conditional SVs. See
also the examples in Section 4 (Figure 1). The decomposition for conditional SVs follows by replacing "conditioning
by intervention" with "conditioning by observation", i.e., by replacing $do(\mathbf{X} = \mathbf{x})$ with $\mathbf{x}$ on the righthand side of the
bar. The decomposition is introduced in Section 3 to assist our illustration of how the different SVs attribute a model's
prediction to the features involved in this prediction in Section 4 for different causal models. Here we also discuss in
which cases (most notably the fork and the confounder) conditional SVs fail to provide an intuitive causal attribution.

Causal chain graphs are introduced as a means to compute causal SVs (whether symmetric or asymmetric) when users
are willing/able to specify a (partial) causal ordering, but not a full-fledged causal model. The asymmetric SVs of [6]
indeed rely on the same information. On top of [6] we offer a formalization in terms of causal chain graphs and show
that, with "conditioning by intervention" instead of "by observation" as in [6], there is no need for asymmetry in the
SVs. Unlike conditional (asymmetric) SVs, causal SVs provide the right intuition in the case of common confounding.

**Reviewer #2**: W.r.t. the novelty in comparison to [6]: asymmetric (conditional) SVs as defined in [6] in some cases
coincide with symmetric or asymmetric (causal) SVs, but are different in general. See also the previous paragraph.

Section 4 aims to illustrate the behavior of the various SVs in simple cases that can be analyzed analytically and then to
argue which is the most intuitive, indeed also linking to psychological literature when appropriate. Here one prominent
theory, dating back to [15], states that humans sample over different possible scenarios to judge causation. Translating
this to a situation in which there are two possible causes, $X_1$ and $X_2$, where it is unknown which one is intervened
upon first, may suggest that the natural interpretation is to consider both options and average over them.

We fully agree that quantifying causal influence is a difficult topic and any method has its weaknesses, but causal
SVs appear to fare better than the reviewer suggests. Discontinuity w.r.t. arrows with zero strength is an issue for the
asymmetric SVs, but not for the symmetric SVs that consider all orderings, not just those consistent with the causal
DAG. After averaging over all these orderings, the indirect effect already does incorporate all possible paths (so we do
not see how or why it needs to be generalized), but of course in the game-specific way inherent to the Shapley value
approach. We will add comments and disclaimers to clarify this and adapt our description of Janzing et al. and related
work as suggested by the reviewer. Our statement 'not every causal query need be identifiable (see e.g., [24])' did not
presume DAGs with all variables observed, but more general causal structures possibly including latent variables.

**Reviewer #3**: W.r.t. the scope, see our answer to Reviewer #1 (third paragraph) and the beginning of Section 5: causal
SVs are generally applicable when a user is willing/able to specify a causal model among the features that are used as
input to the model and when all causal queries are indeed identifiable. Specifying when this is the case is a topic on its
own: we will add more references (see also the supplement). Causal chain graphs are "just" proposed as a practical
approach to handle partial causal knowledge. In causal chain graphs, all causal queries are guaranteed to be identifiable
and can be answered based on the available observational data. These graphs allow for handling cycles, confounders,
etc (see Figure 2). In fact, all examples in Figure 1 are easily translated to causal chain graphs. An illustrative example
for the fork could be predicting hotel occupation ($Y$), based on season ($X_2$) and temperature ($X_1$).

We miss the point the reviewer tries to make w.r.t. counterfactual analysis. As far as we can tell, the counterfactual
question posed by the reviewer (assuming all features are known) can be answered simply by reading off the output of
the model. Our analysis can be interpreted as counterfactual (third rung) reasoning to analyze what the model prediction
would have been had we not known some of the input features (see second paragraph of Section 4). Counterfactual
*explanations* as in e.g. [33] may be improved with similar techniques, but are beyond the scope of the current paper.

Causal relationships are indeed asymmetric, but that does not prevent the causal SVs from being symmetric according
to the standard symmetry axiom for SVs (see the definition in Section 2 and the elaborate discussion in [9], Section 3 in
response to Sundarajan and Najmi, 2019). We chose not to repeat this argumentation, but will add a reference.

Figure 4 is meant to illustrate the difference between the various SVs (asymmetric SVs focus on the root cause, marginal
SVs on the direct effect, symmetric causal SVs consider both), not necessarily to claim that one is always better than
the other. We will extend the supplement with additional empirical analyses, e.g., on (deep) neural networks.

(7) indeed should have been (6). We will fix the other minor issues, also those rightfully indicated by **Reviewer #4**.

[Meta-Review · NeurIPS 2020]

The Shapley value based methodology for explaining a model considers features as players whose coalitions result in establishing the prediction. Formally, the impact of a feature is estimated as the difference between the average Shapley value of the coalitions containing this feature, and that of the coalitions not containing the features. This paper introduces a difference between the direct and indirect effects of a feature, where the difference is whether the out-of-coalition variables are subject or not to an intervention on the target feature. This paper generated an intensive discussion. Some reviewers were enthusiastic about the paper and others found it extremely hard to follow. The discussion somewhat clarified the difference between causal and interventional Shapley values, and the point of replacing the simulated removal of features, with the simulated intervention on features. The terminology needs be simplified: distinguishing between asymmetric conditionnal Shapley and asymmetric interventional Shapley is confusing as the asymmetric interventional approach (this paper) considers the averaging over all permutations. The paper was found very inspirational by some reviewers. Some reviewers regret the experimental illustration to be very sketchy. The authors promised "additional empirical analyses on (deep) NNs", and the AC strongly expect that they will do so if the paper is accepted. The difference with "Asymmetric Shapley Values" should be discussed in more depth and illustrated on more complex cases than the biking. One would lke to see complex cases where it is *not* preferable to give the credit to the root cause only.